# Exploring the Impact of IL-33 Gene Polymorphism (*rs1929992*) on Susceptibility to Chronic Spontaneous Urticaria and Its Association with Serum Interleukin-33 Levels

**DOI:** 10.3390/ijms252413709

**Published:** 2024-12-22

**Authors:** Carmen-Teodora Dobrican-Băruța, Diana Mihaela Deleanu, Mihaela Iancu, Ioana Adriana Muntean, Irena Nedelea, Radu-Gheorghe Bălan, Lucia Maria Procopciuc, Gabriela Adriana Filip

**Affiliations:** 1Department of Allergology and Immunology, “Iuliu Hatieganu” University of Medicine and Pharmacy, 400012 Cluj-Napoca, Romania; dobrican.carmen@umfcluj.ro (C.-T.D.-B.); adriana.muntean@umfcluj.ro (I.A.M.); irena.nedelea@umfcluj.ro (I.N.); balan.radu.gheorghe@elearn.umfcluj.ro (R.-G.B.); 2Allergology Department, “Octavian Fodor” Institute of Gastroenterology and Hepatology, 400162 Cluj-Napoca, Romania; 3Medical Informatics and Biostatistics, Department of Medical Education, “Iuliu Hatieganu” University of Medicine and Pharmacy, 400349 Cluj-Napoca, Romania; 4Department of Biochemistry, “Iuliu Hatieganu” University of Medicine and Pharmacy, 400349 Cluj-Napoca, Romania; lprocopciuc@umfcluj.ro; 5Department of Anatomy, “Iuliu Hatieganu” University of Medicine and Pharmacy, 400006 Cluj-Napoca, Romania; adrianafilip33@yahoo.com

**Keywords:** chronic urticaria, CSU, IL-33, genetic polymorphisms, SNP *rs1929992*, cytokine, autoimmune, susceptibility, personalized medicine

## Abstract

Urticaria is a debilitating skin condition affecting up to 20% of the global population, characterized by erythematous, maculopapular lesions and significant quality of life impairment. This study focused on the role of interleukin 33 (IL-33) and its polymorphisms, particularly SNP *rs1929992*, in chronic spontaneous urticaria (CSU). Using demographic, clinical, and laboratory data from CSU patients and controls, we estimated allele and genotype frequencies, Hardy–Weinberg equilibrium condition, and serum IL-33 levels, using unconditional binomial logistic regression for association analysis. Results revealed that CSU patients had significantly higher frequencies of the minor allele of IL-33 *rs1929992* compared to controls (31.25% vs. 17.35%, *p* = 0.024), and carriers of the GA genotype exhibited increased odds of CSU (adjusted OR = 2.208, *p* ≤ 0.001). Additionally, serum IL-33 levels were markedly elevated in CSU patients, particularly those with the GA genotype. The findings suggest that the IL-33 SNP is associated with an increased susceptibility to CSU, emphasizing its potential as a diagnostic and therapeutic biomarker. This study underscores the genetic and immunological underpinnings of CSU, paving the way for personalized treatment approaches.

## 1. Introduction

Chronic urticaria (CU) is a persistent inflammatory skin condition characterized by erythematous, maculopapular lesions, angioedema, or both, due to mast cell activation and degranulation resulting in histamine and the release of other mediators. While up to 20% of the global population may experience urticaria at some point in their lives, most instances are cases of acute urticaria (AU), which last up to six weeks and are often associated with infections, dietary factors, or medications [1,2]. In contrast, CU, whether spontaneous (CSU) or inducible (CIndU), persists for more than six weeks and typically lasts over a year. CU substantially impacts patients’ quality of life and correlates with psychiatric comorbidities [3] and high healthcare costs [4,5]. CSU differs from chronic inducible urticaria (CIndU) in that the latter presents specific, well-defined triggers that induce symptoms [1,2]. The pathogenesis of CSU involves a series of interconnected events, including autoantibodies, complement activation, and the coagulation cascade [6,7,8,9,10,11]. Diagnosis is clinical, though various tests are conducted to exclude other diagnoses and identify underlying causes in CSU or triggering factors in CIndU [1,2]. Current treatment targets complete response, employing a stepwise approach with second-generation H1 antihistamines, omalizumab, and cyclosporine [1]. Emerging therapeutic strategies focus on targeting mediators, signaling pathways, cytokines [12,13] and receptors of mast cells and other immune cells [14,15,16,17,18,19]. Future research should aim to define disease endotypes and their biomarkers, identify new therapeutic targets, and develop improved therapies, as a significant proportion of patients remain symptomatic with severe and uncontrolled forms of the disease [20,21,22].

IL-33, a member of the IL-1 cytokine family, has been consistently implicated in CSU in previous studies [23,24,25,26,27,28]. Originally discovered in human tissues in 2003 as a nuclear factor of high endothelial venules (NF-HEV), IL-33 has been found to have a three-dimensional folding similar to IL-1 and can induce a type 2 immune response through its receptor, ST2 [29]. The confirmation of IL-33’s identity with NF-HEV and its role as a chromatin-bound nuclear factor were further elucidated [30]. Secreted by various cell types, including endothelial, epithelial, macrophages, fibroblasts, and dendritic cells, IL-33 is released during cellular injury, necrosis, necroptosis, stress, and viral infection, and activates various immune cell types [31,32,33,34]. Although known for its role in allergic reactions, asthma, and parasitic infections, IL-33 has also been suggested to play a role in chronic inflammation due to the complex role of high endothelial venules in lymphocyte activation and mobilization [23].

Given the grounding of autoimmunity in the etiopathogenesis of CSU and its association with various autoimmune diseases as reported in the literature [35,36,37,38,39,40], our research investigated how IL-33, a molecule common to both CSU and autoimmune diseases, has been genetically approached in previous studies. Observing that a specific IL-33 polymorphism, SNP (single nucleotide polymorphism) *rs1929992*, has been associated with susceptibility to various autoimmune diseases [41], raises the critical question of whether this SNP also impacts susceptibility to CSU. This SNP is located within an intronic region of the IL-33 gene, and variations in this region have been linked to altered IL-33 production. Some studies suggest that polymorphisms in IL-33, including *rs1929992*, may influence susceptibility to chronic inflammatory conditions by modulating cytokine levels and the overall immune response. The IL-33 gene is located on chromosome 11 (11q23.1). The SNP *rs1929992* represents the G-A transition located in the IL-33 gene [41].

This study aims to evaluate the prevalence of the IL-33 SNP *rs1929992* in patients with CSU and determine the relationship between this IL-33 SNP and serum IL-33 levels, thus deepening our understanding of the molecular mechanisms involved in this complex condition and identifying potential biomarkers for more accurate diagnostic and personalized therapeutic strategies. The main hypothesis is that IL-33 plays a significant role in the etiopathogenesis of CSU, functioning as an immunological mediator directly involved in inflammation and immune responses [24,25,26,42,43,44]. Through a detailed analysis of demographic and clinical data, allele and genotype frequencies, the Hardy–Weinberg equilibrium, and serum IL-33 levels, while also considering age and sex influences, our hypothesis was tested using robust statistical models, including binomial logistic regression and correction for multiple comparisons, employing the R statistical software, version 4.3.2. This approach has allowed us to elucidate the potential of the IL-33 genetic polymorphism (*rs1929992*) as a predictor of susceptibility to CSU and to deepen the understanding of the underlying etiopathogenic mechanisms. Our findings aim to strengthen both the theoretical framework in the field and the clinical applicability in improving diagnostic and treatment strategies for patients affected by CSU.

## 2. Results

### 2.1. Clinical and Paraclinical Data: Collection and Visualisation

In the present study, sex (*p* = 0.040) and age (*p* = 0.0016) had significantly different distributions in the CSU patients and the non-CSU group (Table 1). As can be seen from Table 1, the baseline IgE level was in the interquartile range of 55.08–355.50 (median 132) in CSU patients; CRP (C-reactive protein) was in the interquartile range of 0.2–0.53 (median 0.30 mg/dL), and 12/48 CSU patients had increased levels of CRP.

### 2.2. Associations of Allele Frequencies of IL-33 rs1929992 Gene Polymorphism with Odds of CSU

The observed genotype distribution of IL-33 *rs1929992* gene polymorphism was consistent with the expected distribution of the Hardy–Weinberg equilibrium in the non-CSU group (*p* = 0.3197) but not in the CSU group (*p* = 0.0016). Table 2 highlights the allele frequencies of IL-33 *rs1929992* gene polymorphism between the groups. We noticed that the frequency of subjects carrying the minor allele of IL-33 *rs1929992* gene polymorphism was significantly higher in the CSU group than in the non-CSU group (31.25% versus 17.35%).

### 2.3. Association Between Genotypes Frequencies of IL-33 rs1929992 Gene Polymorphism and Odds of CSU

The frequencies of genotype distributions of the GG and GA genotypes in the two groups was statistically significant (Table 3); individuals carrying the GA genotype had significantly higher odds of CSU than patients carrying the GG genotype (OR = 3.1, 95% CI: [1.37, 7.19]). Also, we tested the difference in genotype distributions between CSU patients and the non-CSU group adjusting for age and sex in a multivariable logistic model (Table 3). After adjusting for sex and age, association between the *IL-33 rs1929992 gene* polymorphism and odds of CSU remained statistically significant; individuals carrying the heterozygous GA genotype were 2.208 times more likely to develop CSU than individuals carrying the homozygous GG genotype (adjusted *p* ≤ 0:001, adjusted OR = 2:208, 95% CI: 1.548–3.149).

### 2.4. Association Between IL-33 rs1929992 Gene Polymorphism and IL-33 Serum Level Stratified by CSU Patients and Non-CSU Group

The distributions of IL-33 according to cytokine gene polymorphism in CSU patients and the non-CSU group have been presented in Table 4. The median serum level of IL-33 in patients with GA genotype (17.55 pg/mL) was higher than median serum of the GG (15.23 pg/mL) genotype, but the differences were not statistically significant (*p* = 0.5439). Additionally, we noticed that IL-33 levels were higher in the CSU group and lower in the non-CSU group regardless of the GG or GA genotypes of the IL-33 *rs1929992* gene polymorphism (*p* = 0.0000001, Figure 1).

### 2.5. Association of IL-33 rs1929992 Gene Polymorphism with UAS7 and DLQI Scores in CSU Patients

Comparisons of the IL-33 *rs1929992* genotypes among the CSU patients were performed using clinical parameters such as UAS7 scores and DLQI scores (Table 5). Although CSU patients carrying the IL-33 *rs1929992* GG genotype had lower values of baseline UAS7 and DLQI scores than patients carrying the GA genotype, no significant association between UAS7 or DLQI and IL-33 *rs1929992* gene polymorphism was observed (*p* > 0.05).

When we classified disease activity status by the scores of UAS7 as mild or moderate and severe, no significant association was found between the disease activity status and IL-33 *rs1929992* gene polymorphism (Fisher’s exact test, *p* = 0.5451), the frequency distribution of variant GA genotype being similar in patients with severe forms of pruritus (32, 65.6%) or mild/moderate forms of pruritus (16, 56.2%). Also, we found no significant association between IL-33 *rs1929992* gene polymorphism and patients’ quality of life defined using the dermatology life quality index, DLQI, as “moderate impact”, “very important impact”, and “extremely important impact” of disease (Fisher’s exact test, *p* = 0.5451). The frequency distribution of the variant GA genotype was similar in patients with a moderate impact of disease on quality of life (6, 50%), patients with a very important impact of disease (21, 57.1%), and patients with an extremely important impact of disease (21, 71.4%).

## 3. Discussion

This study focuses on the association between the IL-33 *rs1929992* gene polymorphism, serum IL-33 levels, and the risk and severity of CSU, as well as its impact on patients’ quality of life. The genetic polymorphism *rs1929992* of IL-33 has been linked to various autoimmune conditions [43,44,45,46,47], reinforcing its role in the pathogenesis of CSU. Xu et al. [41] initially identified the association between the IL-33 *rs1929992* polymorphism and systemic lupus erythematosus (SLE), which was confirmed by Bagheri-Hosseinabadi et al. [45] in 2023. The comorbidity of CSU with SLE has been extensively documented, demonstrating a potential common pathogenic mechanism [39,48]. Our results, indicating an increased prevalence of the IL-33 polymorphism in CSU patients compared to controls, align with the existing literature, suggesting a strong autoimmune foundation for CSU and underscoring the need to further explore the role of IL-33 polymorphism in this condition.

From another perspective, CSU can be seen as a TH2-type immunological condition, similar to asthma and other allergic diseases [7,9,43]. IL-33, along with IL-25 and TSLP, plays a crucial role as an alarmin in triggering this type of immune response [26]. This aligns with findings from Canoğlu et al. [49] and Rabea et al. [50], who reported the same polymorphism associated with asthma, suggesting its importance in inducing susceptibility to a TH2 immunological profile and in the association of allergic and inflammatory diseases.

Our results indicated an increased incidence of the minor allele among CSU patients, suggesting that genetic variations might influence susceptibility to this condition. The study was based on a robust set of clinical and demographic data and employed appropriate statistical methodologies to test the proposed hypothesis. Adjustments for age and sex were crucial given the initial imbalance between the CSU and non-CSU groups. These corrections ensured a more accurate assessment of the genetic polymorphism’s influence on CSU risk, moving beyond simple correlation to achieve clinically relevant association. Additionally, observations on serum IL-33 levels reinforce the potential of this parameter as a biomarker for assessing and monitoring CSU, although differences between genotypes did not reach statistical significance in our study. This may provide clues on how variations in IL-33 contribute to the inflammatory response and symptomatology in CSU.

Further emphasizing this point, an odds ratio of 3.1, with a *p*-value ≤ 0.001, significantly highlights the increased risk of CSU among individuals with the GA genotype. This strong statistical correlation, coupled with the known role of IL-33 as an alarmin enhancing the immune response during tissue damage and infection, provides a compelling link between genetic predisposition and heightened inflammatory response in CSU. By establishing this connection, we illuminate how the GA genotype may amplify IL-33-mediated signaling pathways, potentially exacerbating the inflammatory response in CSU patients and contributing to disease severity.

The surprising discovery by Guo et al. [51] of an association between stroke and the IL-33 gene polymorphism adds an intriguing element to our understanding. Given that the pathogenesis of CSU partly involves the coagulation cascade [8,41,52], this association, though not fully understood, opens new avenues for research into the deeper mechanisms of the disease. Additionally, the same polymorphism was found in the study by Fang et al. [47] in the context of Henoch–Schönlein Purpura (HSP), an autoimmune, autoinflammatory, and vasculitic condition, adding complexity to this genetic marker. These findings suggest that IL-33 may play a significant role in pathologies involving both inflammation and coagulation disorders and could serve as a starting point for future research in this area.

The array of studies referenced [43,44,45,46,47,48,49,50,51,52] consistently demonstrates how variations in the IL-33 *rs1929992* gene are prevalent across various autoimmune and immunologic diseases, reinforcing our hypothesis that these variations correlate significantly with these conditions. This evidence confirms that the IL-33 *rs1929992* polymorphism can influence serum levels of IL-33, an alarmin known for its crucial role in initiating immune responses and inflammatory cascades. Such findings substantiate the impact of IL-33 in the pathophysiology of immune-mediated and autoinflammatory disorders, including CSU, by linking genetic variations directly to changes in immune system activity.

Because the genotypes in the CSU group show deviations from the Hardy–Weinberg equilibrium (HWE) but those in the non-CSU group do not, this may suggest an association between the genetic polymorphism (IL-33 *rs1929992*) and CSU. The deviation may highlight the effects of natural selection, where certain genotypes are more frequent among patients with CSU due to a dependency relationship with the pathology.

However, the absence of statistical significance between the associations of the IL-33 *rs1929992* polymorphism and clinical UAS7 and DLQI scores is an intriguing component of our study. This finding may suggest that while genetic variants such as the IL-33 *rs1929992* polymorphism play a role in predisposing individuals to CSU, their influence on the severity of symptoms or overall quality of life is not straightforward. This observation underscores the complex interplay between genetics and disease phenotype in CSU, where genetic predisposition does not directly translate to clinical severity.

Further analysis reveals that environmental factors, lifestyle choices, and other biomarkers could have significant impacts on how the disease manifests and progresses in each individual. These factors could potentially overshadow the direct effects of genetic variants on clinical outcomes. This underscores the multifactorial nature of CSU’s clinical expression, where multiple layers of influence—genetic, environmental, and possibly epigenetic—interact to determine the full spectrum of disease presentation and severity. Acknowledging this complexity is crucial for developing more personalized therapeutic approaches that consider both genetic background and external influencing factors.

Despite the valuable insights provided by our study into the genetic landscape of CSU, we must acknowledge its inherent limitations. The nature of our case-control study and the relatively small sample size necessitate a cautious interpretation of the results. While our study identifies an association between the IL-33 *rs1929992* polymorphism and CSU, this correlation—similar to observations in other autoimmune and immunological diseases—does not imply causation or dictate disease severity. This association highlights the complexity of CSU’s etiology and suggests that while the polymorphism may influence disease presence, a comprehensive understanding requires studying additional parameters and subdividing CSU into distinct phenotypes. In addition, to explore the functional impact of the IL-33 *rs1929992* gene polymorphism on IL-33 levels in patients with CSU, we did not control for disease duration or other clinical confounding factors.

In this discussion, we emphasize the potential confounders and highlight the need for careful interpretation of our findings. Further studies with larger cohorts and longitudinal designs are essential to validate our results and deepen our understanding of how these genetic variations affect IL-33 levels and CSU pathogenesis. Such studies could significantly advance our knowledge of cytokine involvement and signaling pathways, potentially leading to optimized therapeutic strategies and improved patient outcomes.

While extensive research has focused on this polymorphism in other autoimmune disorders, our study provides a valuable perspective by investigating these SNPs specifically in CSU within a distinct population—Romania, Europe.

Thus, this study represents an important step in mapping the genetic landscape in CSU and opens promising avenues for targeted and personalized therapies. Continued research into this disease will continue to provide new insights and support the medical community in managing CSU more effectively.

## 4. Materials and Methods

### 4.1. Patient Selection and Study Design

This case-control study was conducted at the Department of Allergology within the Regional Institute of Gastroenterology and Hepatology in Cluj-Napoca, Romania. It involved 50 patients diagnosed with CSU according to the latest international guidelines. CSU is characterized by repeated occurrences of maculopapular rashes and possible angioedema, appearing at least twice weekly for a minimum of six weeks. For comparison, a control group of 50 institutional staff members without any history of urticaria or systemic diseases that could cause urticaria or pruritus were included. The exclusion criteria for CSU patients were closely linked to the diagnostic criteria and excluded any chronic systemic diseases that involve itching or urticaria, including renal, hepatic, psychiatric, or infectious diseases.

The study protocols were approved by the Ethics Committee of the “Iuliu Hatieganu” University of Medicine and Pharmacy, Cluj-Napoca (AVZ270/10 October 2022) and the Regional Institute of Gastroenterology and Hepatology (IRGH) (12637/11 October 2022). All participants gave written informed consent, and basic demographic data, including age and gender, were collected.

The CSU diagnosis was confirmed through patient history and clinical examination, excluding individuals under 18 years or those with other chronic diseases. Blood samples of 10 milliliters were collected from all subjects and divided into four vacutainers: the first two for complete blood count and routine analyses, the third for IL-33 determination via Enzyme Linked Immunosorbent Assay (ELISA) using a commercial kit, and the fourth collected in EDTA for whole blood DNA extraction and molecular study of SNP IL-33 *rs1929992* using Polymerase Chain Reaction—Restriction Fragment Length Polymorphism (PCR-RFLP).

### 4.2. DNA Isolation

Peripheral blood was collected in EDTA tubes. DNA was isolated from leukocytes using the Zymo Research Quick-DNA Miniprep Kit and stored at −20 °C.

### 4.3. PCR and RFLP Analysis

In order to identify the IL-33 IL-33 *G/A* transversion (*rs1929992*), we used the method described by Bassagh (2019) [53] and optimized it in our laboratory. PCR amplification was performed using an iCycler BioRad (Bio-Rad Life Science, Hercules, CA, USA). The reaction mixture contained 20 ng DNA, 0.2 µM forward and reverse primers, 2.0 mM MgCl_2_, 10X Taq polymerase buffer (20 mM Tris-HCl (pH 8.0), 1 mM DTT, 0.1 mM EDTA, 100 mM KCl, 0.5% (*v*/*v*) Nonidet P-40, 0.5% (*v*/*v*) Tween 20, and 50% (*v*/*v*) glycerol), 200 µM each dNTP (dATP, dGTP, dCTP, dTTP), and 0.625 U Taq polymerase enzyme. The sequences of the primers were as follows: forward primer: 5′-GTCATCATCAACTTGGAACCTT-3′ and reverse primer: 5′-CTGTGGAGTGCTTTGCCTTT-3′.

The PCR amplification program was as follows: denaturation for 10 min at 95 °C, followed by 34 cycles of amplification with denaturation for 10 s at 95 °C, annealing of primers for 20 s at 57.6 °C, extension of primers for 27 s at 72 °C, and a final extension for 5 min at 72 °C. The length of the PCR product was 529 bp. After PCR amplification, enzymatic digestion was performed. A 6 µL PCR product was incubated for 3 h at 37 °C with 5 U SspI restriction enzyme. After enzymatic digestion, the wild-type *G* allele produced an undigested fragment of 529 bp, while the mutated *A* allele produced two fragments of 398 bp and 131 bp.

The specificity of amplification and enzymatic digestion was checked using agarose gel electrophoresis, and the fragments were visualized under UV light (Figure 2). The primers were obtained from Kaneka Eurogentec S.A. Biologics Division, Liège, Belgium, and the restriction enzyme was from New England Biolabs, Brüningstr. 50; Geb. B852D-65926 Frankfurt am Main, Germany.

### 4.4. Statistical Analysis

Continuous demographic and clinical variables were summarized using measures of central tendency and dispersion. For normally distributed data, the arithmetic mean and standard deviation were used, while the median and interquartile range (IQR) were applied for non-normally distributed data. Univariate normality was assessed through a combination of descriptive and inferential statistical methods, including descriptive statistics, Q-Q plots, and the Shapiro–Wilk test with Holm correction for multiple comparisons. Qualitative variables were described using frequencies and percentages.

Comparison of demographic and clinical characteristics between CSU and non-CSU groups was performed using Chi-squared test, Fisher’s exact test, Welch *t*-test or Mann–Whitney test for independent samples. The Welch *t*-test was used when the data were approximately normally distributed and the variances of studied characteristics between groups were unequal (heteroscedasticity condition). The Mann–Whitney test was used for comparisons where the data showed deviations from normal distribution.

Differences in allele and genotype frequencies of IL-33 gene polymorphism (*rs1929992*) between CSU and non-CSU groups were compared using the Chi-squared test. The departure from Hardy–Weinberg Equilibrium (HWE) was tested using the exact Chi-squared test from “SNPassoc” R package [54].

The association between IL-33 gene polymorphism (*rs1929992*) and odds of CSU was evaluated by unconditional binomial logistic regression. The effect size of association was described using the unadjusted odds ratio (OR) with 95% confidence interval (95% CI) and adjusted odds ratio (OR) with adjustment for age and sex.

All statistical analysis was performed in R software, version 4.3.2 [55]. The level of statistical significance for all two-sided tests was set at 5%.

## 5. Conclusions

In conclusion, our study underscores the significant role of the IL-33 gene polymorphism (*rs1929992*) in the susceptibility to CSU. Our findings demonstrate that patients with CSU exhibit markedly elevated serum levels of IL-33 compared to control subjects, reinforcing the hypothesis that IL-33 plays a critical role in the etiopathogenesis of CSU.

We specifically explored whether genetic variations in IL-33 contribute to this condition, as has been observed in other autoimmune disorders. Our results reveal a significant association between the frequency of the *rs1929992* allele and the risk of developing CSU. Notably, we observed significant differences in allele and genotype frequencies of the IL-33 *rs1929992* polymorphism between CSU patients and the control group, indicating a heightened susceptibility linked to the minor allele.

This study presents novel evidence that the *rs1929992* polymorphism of the IL-33 gene increases susceptibility to CSU within the Romanian population included in our study. These findings are of considerable importance and originality, as this SNP has not previously been investigated in relation to CSU, and our results highlight meaningful positive associations.

Overall, our research paves the way for future studies to further elucidate the genetic, clinical, and immunological interactions in CSU, ultimately contributing to enhanced clinical management and patient outcomes.

## Figures and Tables

**Figure 1 ijms-25-13709-f001:**
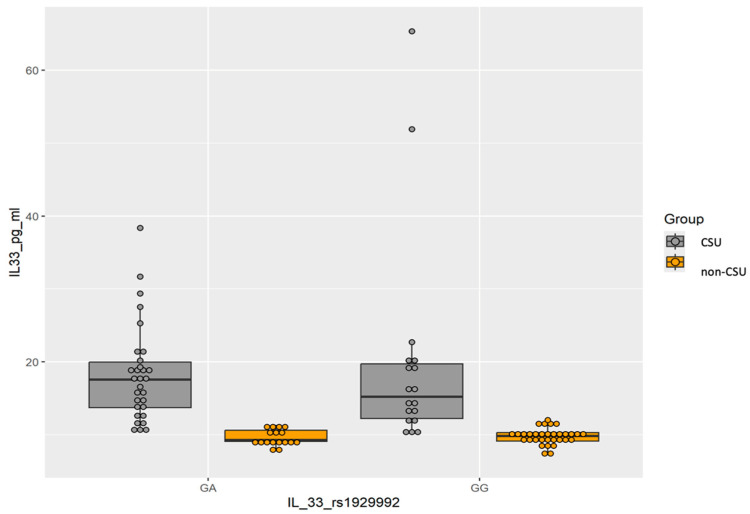
Distribution of IL-33 values in patients carrying GG and GA genotypes of IL-33 *rs1929992* gene polymorphism split by group (CSU group is represented in gray while non-CSU group is represented in yellow). Dot plot shows individual IL-33 levels and box-and-whisker plot shows distribution of data based on median and IQR in studied groups.

**Figure 2 ijms-25-13709-f002:**
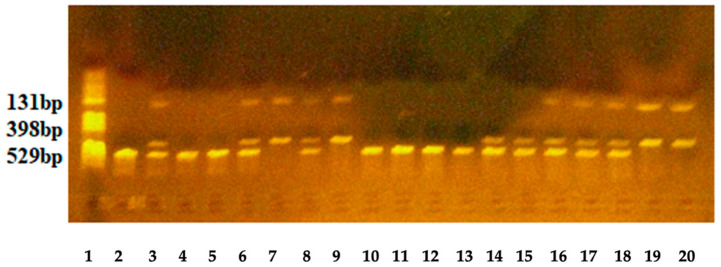
Enzymatic digestion of the 529 bp fragment for IL-33- *rs1929992* identification. Lane 1—pBR HaeIII digest DNA molecular marker V; lanes 2, 4, 5, 10–13—homozygous GG genotype: fragment of 529 bp; lanes 3, 6, 8, 14–18—heterozygous AG genotype: fragments of 529,398 and 131 bp; lanes 7, 9, 19 and 20—homozygous AA genotype: fragment of 398 and 131 bp.

**Table 1 ijms-25-13709-t001:** Comparisons of demographic and clinical characteristics in studied groups.

	Non-CSU Patients (n_1_ = 49)	CSU Patients (n_2_ = 48)	*p*-Values
**Demographic variables**			
Sex [males/females, (% of males)]	22/27 (44.9)	12/36 (35.3)	0.04002 *
Age (years) ^a^	38.7 (14.19)	46.47 (25.0)	0.0016 *
**Paraclinical variables**			
Serum levels of IL-33 (pg/mL) ^b^	9.81 [9.07, 10.40]	16.60 [13.04, 19.98]	<0.0001 *
Baseline IgE level (IU/L) ^b^	ND	132 [55.08, 355.50]	
Baseline blood eosinophils (^x^1000) ^b^	ND	0.36 [0.11, 0.98]	
Baseline C-reactive protein (CRP) (mg/dL) ^b^	ND	0.30 [0.20, 0.53]	
**Treatments**			
Antihistamines only (n, %)	ND	33 (68.75)	
Antihistamines and corticosteroids (n, %)	ND	15 (31.25)	
Atopy (n, %)	ND	13 (27.08)	
Ag.hb.pylori (n, %)	ND	4 (8.33)	

Data were expressed as ^a^ mean = arithmetic mean; SD = sample standard deviation or ^b^ median with IQR = interquartile range [first quartile, third quartile]; ND = not determined; n = number of cases; *p*-values were estimated using Chi-squared test, Welch *t*-test, or Mann–Whitney test. * significant result: *p* < 0.05.

**Table 2 ijms-25-13709-t002:** Allelic frequencies of IL-33 *rs1929992* gene polymorphism and their relationship with odds of CSU.

Gene	rs	Minor Allele	MAF (%)		HWE *p*-Value ^a^	OR [95% CI]	*p*-Value ^b^
			CSU Group	Non-CSU Group			
**IL-33**	** *1929992* **	** *A* **	**31.25**	**17.35**	0.3197	2.17 [1.10, 4.27]	0.024 *

MAF: minor allele frequency; HWE: Hardy–Weinberg Equilibrium; ^a^ calculated in non-CSU group; OR: unadjusted odds ratio; 95% CI: 95% confidence interval; ^b^ estimated from Chi-squared test; statistical significance was achieved if *p* < 0.05. * significant result: *p* < 0.05.

**Table 3 ijms-25-13709-t003:** Genotypic association between selected *IL-33 rs1929992* gene polymorphism and CSU.

Genotypes	Non-CSU Group (n_1_ = 49) n (%)	CSU Group (n_2_ = 48) n (%)	Unadjusted		Adjusted ^a^	
			OR [95% CI]	*p*-Value	OR [95% CI]	*p*-Value
*IL-33 rs1929992*				0.0058 *		0.0133 *
GG	32 (65.3)	18 (37.5)	Reference		Reference	
GA	17 (34.7)	30 (62.5)	3.14 [1.37, 7.19]		2.95 [1.24, 7.03]	
AA	0 (0)	0 (0)	NA		NA	

SNPs: single nucleotide polymorphisms; NA = not available; OR: odds ratio; 95% CI: 95% confidence interval; ^a^ adjusted for age and sex. * significant result: *p* < 0.05.

**Table 4 ijms-25-13709-t004:** Serum levels of IL-33 (pg/mL) according to genetic variations of IL-33 *rs1929992* gene polymorphism.

	IL-33 *rs1929992*	Non-CSU Group (n_1_ = 49)	CSU Group (n_2_ = 48)	*p*-Value
		Median [IQR]	Median [IQR]	
**IL-33 serum level (pg/mL)**	GG (n = 50)	9.85 [9.11; 10.33]	15.23 [12.23; 19.72]	0.0000001 *
	GA (n = 47)	9.29 [9.07; 10.62]	17.55 [13.74; 19.98]	0.0000001 *
	*p*-value	0.7526	0.5439	

IQR = interquartile range [first quartile, third quartile]; frequency of variant genotype (GA) in non-CSU group was 17 (34.7%) and 30 (62.5%) in CSU group; frequency of wild genotype (GG) in non-CSU group was 32 (65.3% and 18 (37.5%) in CSU group; * significant result: *p* < 0.05.

**Table 5 ijms-25-13709-t005:** Distribution of baseline UAS7 and DLQI scores by genotypes of IL-33 *rs1929992* gene polymorphism.

	IL-33 *rs1929992*				*p*-Value
	GG Genotype (n = 18)		GA Genotype (n = 30)		
	Mean (SD)	Range [Min; Max]	Mean (SD)	Range [Min; Max]	
**Baseline UAS7 scores**	27.22 (7.71)	[15; 42]	27.90 (6.80)	[14; 38]	0.7521
**Baseline DLQI scores**	19.06 (6.95)	[8; 30]	21.37 (6.77)	[9; 30]	0.2629

UAS7 = urticaria activity score over 7 days; DLQI = dermatology life quality index.

## Data Availability

All data utilized or generated in this study are presented within the article. Additional questions can be addressed to the corresponding author.

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
