# Peer review of "Exploring the Impact of IL-33 Gene Polymorphism (rs1929992) on Susceptibility to Chronic Spontaneous Urticaria and Its Association with Serum Interleukin-33 Levels"

_ijms, 2024, doi:10.3390/ijms252413709_

Round 1
Reviewer 1 Report (Previous Reviewer 3)
Comments and Suggestions for Authors
This manuscript examines chronic spontaneous urticaria (CSU), a skin condition that affects many people and seriously impacts quality of life. Researchers focused on a genetic variation in the interleukin 33 (IL-33) gene, specifically SNP rs1929992, to determine if it is linked to a higher risk of CSU. They compared the gene variation and blood IL-33 levels in CSU patients and healthy individuals. The results showed that CSU patients were more likely to have the minor allele of this SNP and exhibited higher IL-33 levels, particularly among those with the GA genotype, which doubled the odds of having CSU. These findings suggest that this gene variation might aid in diagnosing CSU and could serve as a target for future treatments, underscoring the genetic and immune factors involved in CSU. This resubmitted manuscript has indeed improved in writing quality, but a few points still need to be addressed:
-
Observational studies like this one have limitations, primarily because conclusions rely on associations rather than causation. In this case, while the study finds a link between IL-33 polymorphisms and chronic spontaneous urticaria (CSU), there may be other factors influencing both the presence of IL-33 polymorphisms and CSU severity that were not accounted for. Without fully controlling for these confounders, it is difficult to determine how direct or exclusive the role of IL-33 is in CSU. The authors should clearly address this limitation in the discussion section.
-
The discussion should also address possible mechanisms through which these gene variations might contribute to CSU development and severity.
-
The authors should carefully check that all instances of allele notation are italicized throughout the manuscript.
Author Response
Please see the attachment.

Reviewer 2 Report (New Reviewer)
Comments and Suggestions for Authors
The manuscript entitled "Exploring the Impact of IL-33 Gene Polymorphism (rs1929992) on Susceptibility to Chronic Spontaneous Urticaria and its Association with Serum Interleukin-33 levels" investigates the association of IL-33 gene polymorphism (rs1929992) with susceptibility to chronic spontaneous urticaria and further its correlation with serum interleukin-33 levels to provide some insight into the pathogenesis of the disease. The paper, though of some scientific interest, raises several concerns with respect to the validity of the data and overall results. Here are some remarks that are important:
1. The Hardy-Weinberg equilibrium in the CSU group was not met (p = 0.0016). Discuss possible reasons, including population stratification or genotyping errors, and the implication for the validity of the study.
2. While IL-33 levels are higher in the CSU group, the lack of statistical significance between genotypes (GG vs. GA) raises questions about the polymorphism's functional impact. Consider using advanced statistical models or subgroup analyses to explore potential confounding factors.
3. Improve table captions and ensure all abbreviations (e.g., UAS7, DLQI) are fully explained for clarity. Enhance Figure 1 by including error bars, a statistical significance marker, and a clear legend.
4. The biological implications of the significant odds ratio are to be elaborated for the GA genotypes and the susceptibility to CSU with OR = 3.1, p ≤ 0.001. Relate findings to broader immunological pathways or previous studies on IL-33.
5. Figure 2. Enzymatic digestion of the 529bp fragment for IL33- rs1929992 identification is fully unclear you have to change with more resolution figure.
6. The conditions for the PCR are very well described, but having decimals of times, like "denaturation for 0.6 sec", is not very conventional. Please clarify whether these were typos or intentional values, as they may confuse readers.
7. The statistical tests were appropriate, but the rationale for the choice of one test over another (Welch t-test vs. Mann-Whitney test) is not explicitly provided. Shortly justify why such tests were used according to data characteristics.
8. Results on the IL-33 rs1929992 genotype association are unequivocal but lack a detailed discussion of nonsignificant findings, such as no association with UAS7 or DLQI scores. This should be discussed to balance the interpretation of results and avoid the impression of selective reporting.
9. Minor typographical errors need to be revised; the English language needs more improvement.
Comments on the Quality of English LanguageThe English language requires further improvement.
Round 2
Reviewer 1 Report (Previous Reviewer 3)
Comments and Suggestions for Authors
The authors have modified the manuscript according to the suggestions.
Reviewer 2 Report (New Reviewer)
Comments and Suggestions for Authors
After thoroughly reviewing the revised manuscript and considering the authors' revisions and responses to the referee's comments, I find that the manuscript has been significantly improved. The authors have effectively addressed the concerns, enhancing their study's clarity and scientific rigor. The revisions have clarified the methodology, improved the presentation of results, and strengthened the discussion and conclusions. Therefore, I believe that the manuscript now meets the standards required for publication in IJMS and I recommend that it be accepted for publication.
Thank you for considering my recommendation.
Comments on the Quality of English LanguageThe English language has undergone improvements and has gained wider acceptance.
This manuscript is a resubmission of an earlier submission. The following is a list of the peer review reports and author responses from that submission.
Round 1
Reviewer 1 Report
Comments and Suggestions for Authors
The manuscript of Dobrican-Baruta et al. investigated a specific IL-33 polymorphism as well as serum IL-33 levels in patients with or without chronic urticaria. Even though, at first glance it looks like an interesting study, it has several flaws, which would need substantial revision.
#1 „CU (…) affecting up to 20% of the global population“ is just not true. Chronic urticaria is much less common (DOI: 10.1111/all.14037). Throughout the manuscript, the authors use urticaria (can be acute or chronic) interchangeably with chronic urticaria (CU). And CU (can be induced or spontaneous) interchangeably with chronic spontaneous urticaria (CSU). They are not the same and it is thus not clear which patients were actually studied. The information about the CU/CSU patients included in this study is sparse. Which tests were undergone to exclude chronic induced urticaria? How long did the patients suffer from their CSU? Was there an autologous serum skin test (ASST) performed to further characterize the CSU patients for the autoimmunity subtype? Because this is one of the major arguments given for looking into the IL-33 polymorphism and IL-33 serum levels.
#2 Also the rs1929992 SNP should be described in the introduction. What is the wild type? Which variants are expected? It is not made clear by the authors which mutation was actually detected: AG or GA are two different mutation outcomes, but they are used interchangeably in this manuscript. Further, the authors state that the mutated A allele would produce two fragments of 398 and 131 bp. In Figure 1, lane 3 and 5 are stated to have a heterozygous AG genotype, however, the 131 bp fragment in line 5 is not visible. And what is this smear in lane 1? Where is the comparable standard DNA ladder? The results of this analysis are thus not clear and might not be reliable, which makes the results of this manuscript questionable.
#3 Table 1: there is missing data in the non-CU patient group. This needs to be measured or stated as missing (e.g. NA). Either use mean or median, preferably median to not confuse the reader.
#4 line 133+134: I would not interpret this data like that. Table 4 shows that IL-33 levels are higher in the CU group and lower in the non-CU group independent of the GG or GA mutation status of IL-33. Table 4 and 5 should be plotted to see every data point (e.g. box plots with single points shown). Split by group, sex and age. Also investigate if IL-33 correlates with age or sex.
#5 Q-Q Plots and Shapiro-Wilk test results are nowhere shown. They should be added as a supplement.
#6 line 256: 5µL of DNA could be anything. Please indicate how many ng were loaded.
Comments on the Quality of English Languagedouble check spelling mistakes:
aestimate->estimate
fregments ->fragments
lupus erythematosus (LES) -> systemic lupus erythematosus (SLE)
UC->CU
Reviewer 2 Report
Comments and Suggestions for Authors
Solid work on identifying polymorphisms in IL33 that associate with CSU risk. Given that many studies have revealed SNPs in the IL33 receptor, ST2/IL1RL1, associate with risk of asthma, it would be useful if authors also tested for this in their CSU samples. They can use the SNPs that have been described. This will make their story complete. Authors should also measure serum soluble ST2 to see if it tracks with CSU risk.
Author Response
Dear Reviewer,
Thank you very much for your insightful and valuable suggestion regarding the assessment of polymorphisms in the ST2/IL1RL1 receptor and the measurement of serum soluble ST2 in our chronic spontaneous urticaria (CSU) cohort. We agree that this would indeed enrich and broaden the scope of our study, potentially offering a more comprehensive understanding of the relationship between IL33 and its receptor in CSU.
However, due to financial constraints, we were unable to measure serum soluble ST2 and the SNPs in this current study. The high cost of the necessary kits and the logistical challenges of recalling all participants (patients and controls) for new blood sample collection would require significantly more time than is available during this revision process. Unfortunately, the 10-day revision window does not allow us to perform these additional analyses.
Moreover, these specific analyses were not part of our research's original aims, and as such, incorporating them at this stage would extend beyond the scope of the current study. Nevertheless, we acknowledge the importance of this suggestion and will certainly consider it for future research.
We appreciate your recommendation and will consider it in our future work.
With respect and gratitude,
Carmen-Teodora Dobrican-Băruța and co-authors
Reviewer 3 Report
Comments and Suggestions for Authors
This manuscript investigates the clinical correlation between genetic variants, particularly SNP rs1929992 of interleukin-33 (IL-33), and chronic spontaneous urticaria. By analyzing data from CU patients and healthy controls, the study examines allele and genotype frequencies, genetic equilibrium, and serum IL-33 levels. The findings suggest that CU patients are more likely to carry the IL-33 rs1929992 minor allele, indicating a higher susceptibility to CU. Additionally, elevated serum IL-33 levels were observed in patients, especially those with specific genotypes. These results propose that the IL-33 SNP rs1929992 may play a key role in CU susceptibility and could serve as a valuable diagnostic and therapeutic biomarker. The study highlights the significance of genetic and immunological factors in CU, opening avenues for personalized treatment approaches. Below are the major concerns regarding this manuscript:
-
The study on IL-33 polymorphisms and chronic urticaria, while insightful, appears to be primarily observational in nature, which introduces several potential limitations. For instance, the study may not have fully accounted for all possible confounding variables that could influence both the presence of IL-33 polymorphisms and the severity of chronic urticaria. Without controlling for these confounders, the results may be affected by external factors that were not considered. Therefore, further experimental studies would be valuable to confirm the role of IL-33 polymorphisms on IL-33 function and to better understand the underlying mechanisms.
-
The writing is somewhat rough, making the manuscript difficult to understand. A more detailed and logically structured description would benefit the readers.
-
In lines 93-94, the authors state: "when we tested the difference in genotype distributions between CU patients and the non-CU group, age and sex were adjusted." However, there is no explanation in the manuscript on how this adjustment was performed.
-
Several abbreviations, such as CRP, are not spelled out in full when first introduced. Please double-check all abbreviations and provide their full names upon first use. Additionally, the abbreviations 'CU' and 'CSU' appear to be used interchangeably; are they considered synonyms? If so, the use of these abbreviations should be consistent throughout the manuscript. If not, the difference between them should be clearly defined and explained in the manuscript.
-
In the introduction, the authors should provide more information regarding rs1929992 and the correlation between IL-33 and CU.
Comments on the Quality of English Language
Minor editing of English language required.
Author Response
Dear Reviewer,
Thank you for your valuable feedback. I will address each of your comments individually below:
- "The study on IL-33 polymorphisms and chronic urticaria, while insightful, appears to be primarily observational in nature, which introduces several potential limitations. For instance, the study may not have fully accounted for all possible confounding variables that could influence both the presence of IL-33 polymorphisms and the severity of chronic urticaria. Without controlling for these confounders, the results may be affected by external factors that were not considered. Therefore, further experimental studies would be valuable to confirm the role of IL-33 polymorphisms on IL-33 function and to better understand the underlying mechanisms."
Response: Thank you for highlighting this important aspect. We agree that our study is observational and, as such, has limitations in controlling for all potential confounders. While we have adjusted for key variables such as age and sex, we acknowledge that additional confounders could play a role in both IL-33 polymorphisms and chronic urticaria severity. As suggested, further experimental studies, including functional assays, will be valuable to clarify the mechanistic relationship between IL-33 polymorphisms and their effect on IL-33 function. We have addressed this limitation in the manuscript's discussion section and highlighted the need for future research in this area.
- „The writing is somewhat rough, making the manuscript difficult to understand. A more detailed and logically structured description would benefit the readers.”
Response: Thank you for your constructive feedback regarding the clarity of the writing. We have thoroughly revised the manuscript to improve its readability and logical flow. We believe these revisions enhance the clarity and presentation of our findings, making the manuscript easier for readers to follow.
- „In lines 93-94, the authors state: 'when we tested the difference in genotype distributions between CU patients and the non-CU group, age and sex were adjusted.' However, there is no explanation in the manuscript on how this adjustment was performed.”
Answer: Thank for your suggestion. In the Statistical Analysis section, we stated that “the association between IL-33 gene polymorphism (rs1929992) and odds of CU was evaluated by unconditional binomial logistic regression. The effect size of association was described using the unadjusted odds ratio (OR) with 95% confidence interval (95% CI) and adjusted odds ratio (OR) with adjustment for age and sex”. So, when we estimated the effect of the studied gene polymorphism on risk of CU, we tested a multivariable logistic model including IL33 rs1929992 gene polymorphism as main explicative variable and age and sex as covariates.
To eliminate any confusion, we removed this statement next to the paragraph describing the results of Table 3, as follows: “Also, we tested the difference in genotype distributions between CSU patients and non-CSU group adjusting for age and sex in a multivariable logistic model (Table 3).” (please see lines 133-153).
- "Several abbreviations, such as CRP, are not spelled out in full when first introduced. Please double-check all abbreviations and provide their full names upon first use. Additionally, the abbreviations 'CU' and 'CSU' appear to be used interchangeably; are they considered synonyms? If so, the use of these abbreviations should be consistent throughout the manuscript. If not, the difference between them should be clearly defined and explained in the manuscript."
Response: We apologize for the oversight regarding the abbreviations. We have carefully reviewed the manuscript and ensured that all abbreviations, including CRP (C-reactive protein), are spelled out in full upon their first mention. Regarding the use of 'CU' and 'CSU', you are correct that they are not interchangeable. Our study specifically focuses on chronic spontaneous urticaria (CSU), and we have now made the distinction clearer throughout the manuscript, ensuring consistent use of terminology. We appreciate your observation, which has helped improve the manuscript’s clarity.
- "In the introduction, the authors should provide more information regarding rs1929992 and the correlation between IL-33 and CU."
Response: Thank you for your suggestion. We have expanded the introduction to include a more detailed description of the rs1929992 SNP and its potential role in the pathophysiology of chronic spontaneous urticaria (CSU). We have also added relevant references to further explain the relationship between IL-33 polymorphisms and the development of CSU, which we believe enhances the background and context for our study.
Thank you once again for your helpful and constructive feedback. We believe the revisions we have made have strengthened the manuscript and addressed the concerns you raised.
Respectfully,
Carmen-Teodora Dobrican-Băruța on behalf of all authors

Round 2
Reviewer 1 Report
Comments and Suggestions for Authors
My comments were adequately adressed and the manuscript improved.
Author Response
Dear Reviewer,
Thank you very much for your help and guidance in shaping the manuscript into a much improved version through your insightful comments.
Warm regards,
Carmen-Teodora Dobrican-Băruța
Reviewer 2 Report
Comments and Suggestions for Authors
no additional comments
Author Response

(The authors gave the same response as above.)

Reviewer 3 Report
Comments and Suggestions for Authors
This revised version of the manuscript has improved the data description. However, the major concern is that the manuscript remains primarily observational in nature, with descriptive results, which limits its scientific significance. Additionally, the title “Exploring the Impact of IL-33 Gene Polymorphism (rs1929992) on Susceptibility to Chronic Urticaria and its Association with Serum Interleukin-33 Levels” should be corrected to “Exploring the Impact of IL-33 Gene Polymorphism (rs1929992) on Susceptibility to Chronic Spontaneous Urticaria and its Association with Serum Interleukin-33 Levels,” as rs1929992 is associated with chronic spontaneous urticaria (CSU).
Comments on the Quality of English LanguageMinor English editing could be helpful.
Author Response
Dear Reviewer,
Thank you very much for your feedback. Regarding your comment on the title, I kindly ask you to review it once more, as there appears to be no difference between the title we originally proposed and the one suggested in your comment.
Additionally, to clarify the significance of this study: based on our research, the IL-33 gene polymorphism rs1929992 has not previously been investigated in the context of chronic spontaneous urticaria (CSU). It has only been explored in relation to other autoimmune conditions, as outlined in both our introduction and discussion sections. Given CSU’s autoimmune etiology and the known involvement of IL-33 in its pathogenesis (which we have documented in our manuscript), we aimed to examine whether the rs1929992 SNP of IL-33 might also hold relevance for CSU. Our results, which we sought to present concisely, indeed indicate a potential association, as emphasized in our discussion and conclusion.
Thank you once again for your insights and guidance.
Best regards,
Carmen-Teodora Dobrican-Băruța
Round 3
Reviewer 3 Report
Comments and Suggestions for Authors
The "Chronic Utricaria" in the title of the manuscript, should be modified as " chronic spontaneous urticaria", because rs1929992 is associated with chronic spontaneous urticaria (CSU) but not chronic utricaria.
Please double check the title of manuscript. The manuscirpt title “Exploring the Impact of IL-33 Gene Polymorphism (rs1929992) on Susceptibility to Chronic Urticaria and its Association with Serum Interleukin-33 Levels” should be corrected to “Exploring the Impact of IL-33 Gene Polymorphism (rs1929992) on Susceptibility to Chronic Spontaneous Urticaria and its Association with Serum Interleukin-33 Levels,” as rs1929992 is associated with chronic spontaneous urticaria (CSU).
Author Response
Dear Reviewer,
Thank you very much for your valuable suggestion: you are absolutely right. We sincerely appreciate your attention to detail and your commitment to the scientific accuracy of our work. Your input has been instrumental in ensuring that our manuscript accurately reflects the association between the IL-33 gene polymorphism and chronic spontaneous urticaria.
We have made the necessary changes to the title, and we have also revised the conclusions to clearly highlight the connection between our findings and results, ensuring that they support each other. Additionally, we have implemented improvements to the study design. These modifications are highlighted in blue for your convenience.
We are profoundly grateful for your guidance. Your assistance not only enhances our current research but also contributes to the broader scientific discourse on this subject.
We hope that these modifications meet your expectations, and we kindly ask you to specify any further requirements or suggestions you may have, as we are eager to address them.
Thank you once again for your crucial contribution!
Best regards,
Carmen-Teodora Dobrican-Băruța
